# Influence of Seed Coat Integrity on the Response of Pepper Seeds to Dielectric Barrier Discharge Plasma Treatment

**DOI:** 10.3390/plants14131938

**Published:** 2025-06-24

**Authors:** Chanyanuch Sriruksa, Choncharoen Sawangrat, Sakon Sansongsiri, Dheerawan Boonyawan, Sa-nguansak Thanapornpoonpong

**Affiliations:** 1Department of Plant and Soil Science, Faculty of Agriculture, Chiang Mai University, Chiang Mai 50200, Thailand; thanyalak_sr@cmu.ac.th; 2Department of Industrial Engineering, Faculty of Engineering, Chiang Mai University, Chiang Mai 50200, Thailand; choncharoen.s@cmu.ac.th; 3Plasma and Beam Physics Research Facility, Faculty of Science, Chiang Mai University, Chiang Mai 50200, Thailand; sakon.sa@cmu.ac.th (S.S.); dheerawan.b@cmu.ac.th (D.B.)

**Keywords:** seed coat conditions, dielectric barrier discharge plasma, seed quality, pepper

## Abstract

This study investigated the response of pepper seeds with varying seed coat conditions (SCs) to dielectric barrier discharge plasma treatment (PT). The experimental design was a split plot with three replications. The primary plot factor was the SC (normal seeds [NMS], nicking at the hilum part [NHP], and removed seed coat [RSC]), while the subplot factor was the plasma exposure time (0.4–2.0 s/cm), including a control, to determine the effects on seed viability, germination, and vigor. The results indicate that NMS seeds exhibit the highest performance in terms of seed viability. The NMS and NHP had statistically significantly higher seed germination, electrical conductivity, radical emergence, and germination index at 14 days after sowing, and the shoot length measured longer than RSC. Plasma exposure at 1.2 s/cm improved germination and vigor, whereas 2.0 s/cm exposure significantly decreased seed viability and increased the number of abnormal seedlings. The interaction between SC and PT significantly affected seedling abnormalities, with RSC seeds being more vulnerable to damage under prolonged exposure. These findings highlight the crucial role of seed coat integrity in maintaining seed quality and suggest that carefully controlled PT can be a promising and sustainable method to enhance pepper seed performance.

## 1. Introduction

Peppers are an essential ingredient in several dishes worldwide, and Thailand’s seed industry holds a prominent position on the world stage, ranking fifteenth globally and as the third-largest vegetable seed exporter in Asia [1]. The country is known for producing high-quality pepper seeds for export to various countries worldwide. However, the pepper seed export industry faces several challenges, with seed quality being a crucial factor that directly impacts market access, crop yields, and overall competitiveness. Therefore, maintaining seed quality throughout the export process is important for ensuring that the seeds meet international standards. One such practice is seed enhancement technology, which aims to improve and enhance seed quality after harvesting.

These technologies include seed priming, seed stratification, and seed scarification. Seed priming is a seed treatment that activates the initial stages of germination without allowing the seed to fully germinate; it involves controlled hydration and dehydration cycles to bring seeds to a uniform stage of readiness for planting [2,3]. Seed stratification is a technique in which seeds are exposed to specific temperatures and moisture conditions to break dormancy [4]. Seed scarification is a process used to enhance seed germination by weakening or breaking the seed coat, allowing water and gases to penetrate more easily [5,6]. These technologies can significantly enhance seed germination by addressing dormancy and optimizing the environmental conditions; however, each method has specific requirements and potential drawbacks. For instance, primed seeds typically have a shorter shelf life than unprimed seeds, usually not exceeding 6 months to a year [2]. The risk of over-scarification damaging the seed coat can result in infection or reduced viability. Chemical handling risks in acid scarification require careful handling to avoid accidents and ensure safe disposal [4]. Owing to these limitations, research is currently being conducted on seed conditioning methods, one of which is the use of plasma.

Plasma is an ionized gas composed of ions, excited particles, free radicals, and electrons. A key characteristic of plasma is its ability to emit electromagnetic radiation, including ultraviolet (UV) light. This emission occurs due to the de-excitation of excited species within the plasma and the recombination of ions and electrons [7]. These properties make plasma treatment suitable for emerging as a promising alternative to traditional seed conditioning methods. Plasma treatment offers a range of benefits, including improved germination rate, enhanced seedling vigor, and increased resistance to environmental stresses. Unlike traditional methods that may involve harsh chemicals or physical treatments, plasma treatment (PT) is a relatively gentle and environmentally friendly approach [8,9,10,11,12].

Plasma technology in agriculture is primarily based on non-thermal plasma, which operates at temperatures close to room temperature, making it suitable for treating biological materials without causing thermal damage [13,14]. The primary types of plasma used in agriculture include dielectric barrier discharge (DBD), atmospheric pressure jets (APPJs), corona discharge, spark discharge, and underwater discharge [15]. DBD plasma is the most popular type of plasma used in agriculture because it offers a unique combination of efficiency, scalability, and versatility, making it suitable for a wide range of applications from surface modification to environmental remediation. The ability to generate non-equilibrium plasma conditions at low temperatures enhances its utility in various industrial and research settings [16,17].

The application of DBD plasma for suitable durations has been shown to positively influence seed germination in diverse species, such as kangkong [18], wheat [19,20], rice [21,22], quinoa [23], lettuce [24], and Chinese kale [25], by inducing alterations in physical and biochemical factors that improve seed quality. For instance, the germination percentage of kangkong seeds increased by DBD PT at 15 min with a power of 5.5 kV can be improved; the higher germination rate was at a value of 96.67% compared to the control treatment (86.00%) [18]. DBD plasma treatment of black gram seeds enhanced the formation of nitrogen complexes in the seed coat. The promoted nitrogen conversion increases protein content in the seed coat, enhancing germination rates and seed vigor. This likely results from interactions with reactive plasma species during discharge, which also accelerate imbibition. Nitrogen complexes accumulate by diffusing and absorbing onto the rough, waxy seed coat, where they become partially trapped. As water uptake rises, reactive nitrogen species adsorb onto the uneven seed coat surface and diffuse through its thin layers, further increasing nitrogen content in both the seed and seed coat [26]. Moreover, plasma treatment may stimulate biochemical pathways critical for seed germination. During this process, key biochemical activities are activated, resulting in elevated levels of gibberellic acid (GA_3_) and hydrolytic enzymes that facilitate the mobilization of stored nutrients in the endosperm [27].

In the context of research on plasma technology’s effects on seed quality, relatively few studies have examined the physical properties of seeds, such as surface damage and structural changes. For instance, prolonged exposure of mung bean seeds to DBD plasma can result in excessive etching or erosion of the seed coat. This may result in cracks, shrinkage, or deformation of the seed epidermis; such damage compromises the protective function of the seed coat, making seeds more vulnerable to environmental stressors such as pathogens and dehydration [28]. Overexposure results in reduced germination, and research on Chinese kale (*Brassica oleracea* var. *alboglabra*) showed that exposure beyond optimal durations led to lower germination percentages and poor seedling development [25]. In addition to the loss of seed coat integrity, PT can remove protective wax layers or alter lipid components in the seed coat, resulting in increased permeability. While this enhances water uptake, it may also make seeds more susceptible to water loss or microbial invasion under unfavorable conditions [13].

SCs play important structural and functional roles in shielding embryos from mechanical damage, pathogen dedication, extreme temperatures, and water absorption gas exchange during germination through structures such as the micropyle and hilum [13,29,30]. The hilum is the scar on the seed coat, marking the point at which the seed was attached to the ovary wall via the funicle during development. The hilum is a critical seed structure that supports water absorption, gas exchange, and nutrient flow while influencing seed dormancy, germination efficiency, and overall seed quality [29,30,31,32,33,34].

However, research focusing on the physical properties of seeds subjected to PT remains limited. Therefore, further investigation is necessary to elucidate the mechanisms underlying seed quality improvement. It is hypothesized that the structure of the seed coat influences the internal seed part response to plasma treatment, which, in turn, affects seed germination and seedling development.

The aim of this study was to determine how pepper seeds with different SCs respond to PT by evaluating seed viability, seedling development, germination rates, and seed vigor.

## 2. Results

The optical emission spectroscopy (OES) spectrum shows the emission intensity (a.u.) as a function of wavelength (nm) (Figure 1). This spectrum identifies the species present in the plasma based on its characteristics. The peaks at around 337, 357, and 380 nm correspond to molecular nitrogen (N_2_) and ionized nitrogen (N_2_^+^). These peaks suggest that nitrogen plays a crucial role in plasma discharge. A strong peak appears at 310 nm (emission intensity value 64,431 a.u.), which corresponds to OH emission. The presence of OH radicals indicates water vapor or hydroxyl groups in the plasma, often resulting from residual moisture. The cluster of peaks between 200 and 280 nm is attributed to nitric oxide (NO) emissions (Figure 1a). Argon (Ar) emission lines appear in the 700–900 nm range and are identified as Ar emissions (e.g., 750, 763, and 811 nm). The oxygen (O) peaks around 777, 794, and 801 nm correspond to atomic oxygen (O) emissions [35,36] (Figure 1b).

Table 1 shows a tetrazolium-stained experiment that revealed at least three patterns (Figure 6) in the tested seeds. The patterns indicate significant differences in the viability under each SC. After removing the seed coat, we found that the removed seed coat (RSC) had a decreased percentage of viable and vigorous (VG) patterns compared with normal seed (NMS) and nicking at the hilum (NHP). It significantly decreased to 94% (*p* ≤ 0.05), whereas the VG of NMS was 97%, and the NHP was 95%. Moreover, the results of viable and non-vigorous (NVG) patterns showed that RSC had a statistic significantly increased percentage of NVG compared with NMS and NHP, where the representative patterns were cotyledons displayed colorless (CDC, 0.66%), hypocotyl displayed colorless (HDC, 1.44%), and radicle displayed colorless (RDC, 0.56%). Our results of non-viable patterns showed a significant increase in seed nicking in the hilum part, which was 4%, while RSC and NMS were 3% and 2%, respectively.

The duration of DBD PT significantly affected the seed viability. Specifically, VG decreased to 88% following 2.0 s/cm plasma exposure, compared with 99% in the control group. Furthermore, the 2.0 s/cm PT resulted in increased NVG incidence as well as a higher proportion of non-viable seeds.

The NMS and NHP seeds exhibited the highest G^1st^ and G^Final^ values (55% and 82% and 55% and 74%, respectively), whereas the RSC seeds had significantly lower germination (15% and 37%).

The electrical conductivity (EC) values varied significantly among the seed coat types, with NHP seeds showing the highest EC (741.48 µS cm^−1^ g^−1^), indicating more electrolyte leakage. Radicle emergence (RE) followed a similar trend, with NMS and NHP showing better performance than RSC.

The RE was significantly greater in NMS and NHP seeds (55.56% and 55.33%, respectively) than in the RSC seeds (23.00%). This trend aligns with the G data, indicating better seedling establishment for the NMS and NHP types.

The NHP seeds exhibited statistically a significantly higher germination index (GI) (*p* ≤ 0.05), with the values recorded at 7 and 14 days after sowing (7.56 and 29.42, respectively). In contrast, the RSC seeds showed that GI values were 1.70 and 11.84, respectively, indicating slower and less uniform germination.

The shoot and root lengths were significantly greater in NMS and NHP than in RSC. Specifically, NMS produced shoots with an average length of 3.67 cm and roots with an average length of 2.09 cm, whereas RSC seeds exhibited a reduced shoot length of 2.71 cm. These results suggest that SC integrity plays a crucial role in promoting seedling vigor.

The data in Table 2 demonstrate that SC and PT strongly influence seed performance. NMS and NHP seeds performed better than RSC, and plasma exposure at 1.2 s/cm further enhanced germination speed and seedling vigor. However, plasma overexposure (2.0 s/cm) adversely affected seed viability.

The effects of SCs on abnormal seedling (AS) types are summarized in Table 3. SC treatment showed significant effects on seedling rot (SR), seedling decay as a result of secondary infection (SDSI), and the primary root is trapped in the seed coat (PTS). Seeds from the RSC group showed a significantly higher percentage of SDSI (2.56%) compared with the NMS and NHP groups, whereas the NHP seeds showed significantly higher SR (1.22%) and PTS (7.44%) than the NMS and RSC groups.

For the PT factor, significant differences were observed across multiple AS types. PT at 2.0 s/cm resulted in the highest SR percentage (2.22%), which was significantly increased compared with the control and most other treatments. Furthermore, PTs had significant effects on primary root missing (PM) and PTS, particularly at higher exposure durations (2.0 s/cm).

Furthermore, the percentage of dead seeds varied significantly between the seed coat types and PTs. Seeds from the RSC group showed the highest number of dead seeds (46%) compared with the NMS (10%) and NHP (8%) groups. Increasing the PT time to 2.0 s/cm also significantly increased the dead seed rate to 29%.

The interaction between SC and PT was significant for SR, PTS, PM, and dead seeds, suggesting that the effect of plasma exposure was dependent on the SC for these traits.

The interaction between the SC and PT duration is presented in Figure 2. RSC seeds had consistently lower normal seedling rates than NMS and NHP across all exposure PTs. PT had a negative impact on RSC seedlings, as the percentage of normal seedlings significantly decreased with an overexposure time of 1.2 s/cm (Figure 2a). RSC seeds showed a notable increase in AS rates with prolonged PT, especially beyond 0.8 s/cm. NMS and NHP seeds maintained low AS percentages (Figure 2b). A sharp increase in the percentage of dead seeds was observed in the RSC group as the plasma exposure duration increased. In contrast, the NMS and NHP seeds maintained consistently low dead seed rates regardless of PT, highlighting the heightened vulnerability of the RSC seeds (Figure 2c).

## 3. Discussion

The results clearly show that SC and PT significantly influence seed quality, with notable interactions between them. The differences observed in seed viability, germination, and vigor among the three SCs (NMS, NHP, and RSC) reflect the crucial role of physical and physiological seed structures in regulating seed performance. The NMS and seeds with mechanical scarification by NHP consistently outperformed the seeds with RSC across most parameters, including seed viability, seed germination, and seed vigor.

The TZ test revealed a strong influence of SC on the seed metabolic activity. The NMS seeds showed the highest proportion of vigorous tissues (VG), indicating superior metabolic activity and intact viability. NHP seeds, which had a localized disruption at the hilum, showed slightly reduced, but still acceptable viability, suggesting that minor physical modification did not extensively compromise the metabolic function. The RSC showed a markedly higher proportion of colorless tissues in the CDC, HDC, and RDC, indicating reduced viability. This suggests that the absence of the seed coat compromises the metabolic integrity during imbibition and early germination. The seed coat plays a crucial role in regulating oxidative stress and enzymatic activity during early seedling development [37]. Its removal exposes the inner tissues to pathogen invasion and oxidative damage, accelerating cell death [38]. This may explain the observed low EC readings, as the dead tissues no longer actively release measurable ions.

Germination data further supports the important role of the seed coat structure. NMS showed the highest values for G^final^, GI, and G^1st^, indicating that the intact physical structure supports water regulation, enzyme activation, and synchronization of germinative events. NHP seeds performed comparably well, showing that controlled nicking can facilitate water uptake and reduce dormancy barriers without compromising the physiological integrity. This is in agreement with the findings of Jing et al. [39], who noted that slight mechanical treatments can enhance water permeability and break seed dormancy when carefully applied.

In contrast, RSC seeds showed significantly reduced G^final^, GI, and seedling growth traits. The mechanical removal of seed coats likely disturbed water regulation, increased susceptibility to desiccation, and impaired enzyme activation. Kovalski et al. [40] found that intact seed coats controlled water uptake and reduced imbibitional damage, thereby enhancing germination and vigor. While the EC values appeared lower, this paradoxically indicates impaired ion transport mechanisms rather than membrane integrity; this indicates a dysfunctional cellular system [41]. The EC values were lowest in the RSC seeds, indicating better membrane stability in the NMS and NHP seeds. However, despite the high EC values of NMS and NHP, their viability remained superior to that of RSC, suggesting that membrane leakage was not the sole determinant of seed vigor [42].

AS observations further highlighted the detrimental effects of the seed structure. The RSC seeds showed the highest incidence of abnormalities such as SR, SDSI, and PSS. This reflects compromised structural protection, enhanced pathogen penetration, and poor physiological support during early seedling establishment. These results demonstrate the role of the seed coat as a physical and biochemical barrier against microbial invasion [38]. In contrast, NMS seedlings showed the lowest abnormality rates, whereas NHP showed moderate abnormalities primarily related to hilum-associated root emergence.

The application of plasma significantly influenced seed germination depending on the exposure time. The 1.2 s/cm treatment yielded the best results, enhancing G^1st^ (50%), RE (54%), and GI (26.03 at 14 days) without causing cellular damage. These outcomes agree with those of Randeniya and de Groot [42], who emphasized that moderate plasma exposure improves water uptake, enzyme activity, and seed vigor by modifying surface properties. Short plasma exposure increased seed coat hydrophilicity, promoted water absorption, and improved oxygen availability, all of which contributed to faster and more uniform germination [43]. This mechanism has been demonstrated in several crop species, such as rapeseed (*Brassica napus* L.) [44], wheat [19], and tomato [45]. In contrast to prolonged exposure time, plasma can disrupt seed cell membranes, leading to the leakage of intracellular compounds and a consequent increase in the EC of the surrounding water. This elevated EC serves as an indicator of the role of seed vigor, excessive EC resulting from overexposure can hinder germination and growth [29].

However, at an exposure time of 2.0 s/cm, overexposure results in higher AS incidence (Table 3) and reduced viability (Table 1). This effect is likely attributable to oxidative stress induced by excessive reactive oxygen species (ROS), which can damage membrane lipids, proteins, and nucleic acids, impairing cellular functions and reducing germination potential [46].

The highest rates of seedling abnormalities, particularly SR and PM, were observed in seeds exposed to 2.0 s/cm plasma exposure. These findings are consistent with the results of the TZ test, which revealed that ASs were often associated with seeds showing incomplete or irregular staining patterns (Table 1).

Moreover, specific TZ staining patterns were closely correlated with these types of seedling abnormalities. For instance, CDC was frequently associated with cotyledon deformation (CD), whereas HCD was linked to a broader range of abnormalities, including seedling deformation (SD), SR, SDPI, SDSI, and, in some cases, hypocotyl deformation (HD). Likewise, RCD was commonly related to physiological defects such as PTS, PSS, and PM. These correlations suggest that the type and severity of seedling abnormalities can be effectively predicted based on specific TZ staining patterns.

This suggests that high plasma doses can induce latent cellular injuries, weaken structural tissue development, and compromise microbial resistance. Interestingly, moderate PTs (0.4–1.2 s/cm) result in fewer abnormalities than the control treatment, suggesting a sterilizing effect at lower dosages; this aligns with Sivachandiran and Khacef [47], who proposed plasma as a dual-purpose tool for seed disinfection and physiological priming when applied appropriately.

The NMS and NHP responded more positively to plasma exposure, particularly at moderate durations (0.4–1.2 s/cm), compared with the RSC seeds. This suggests that plasma penetration and surface activation are more effective in seeds with permeable outer layers; this is because of changes in the seed structure. After PT, the plasma etches the seed coat and makes the seed coat thinner, with roughness and widening cracks around the surface [18]. This allows better contact with water, allowing water and oxygen from the outside to be better absorbed into the seed. At the same time, free radicals that pass into the seed will signal biochemical changes by promoting the synthesis of the gibberellin hormones and breaking down the abscisic acid, resulting in higher seed germination (Figure 2a) and faster germination [9,13,48] such as RE and GI in Table 2.

When plasma was applied to NHP, a more variable outcome was observed. Although the germination rate increased compared with that of NMS, there was a modest rise in the number of abnormal seedlings (Figure 2a) and dead seeds (Figure 2c). The hilum is a physiologically active zone, and its exposure can allow reactive plasma species (e.g., ozone and nitric oxide) to penetrate the internal tissues. This may cause localized oxidative stress, damage embryonic structures, or disrupt enzyme systems [37]. As demonstrated in the TZ test, seeds with a wider hilum area often exhibited partial or uneven staining, particularly in the radicle, which appeared colorless or weakly stained (Table 1).

The combination of NHP and PT at 2.0 s/cm resulted in an increased percentage of ASs. In particular, the AS type is SR. This effect from overexposure or improper plasma conditions can weaken seedling tissues, making them more susceptible to rot. Furthermore, if not well controlled, reactive oxygen and nitrogen species generated by plasma can induce oxidative stress, potentially damaging seed cells and impairing their ability to resist pathogens, resulting in SR [44,49].

The RSC showed heightened sensitivity to PT, as evidenced by the increased rates of AS (Figure 2b) and dead seeds (Figure 2c). This heightened sensitivity can be attributed to the inherent physical and biochemical properties of RSC seeds, such as higher moisture content [13] and reduced structural rigidity, which render them more susceptible to plasma-induced damage [50]. PT can alter the seed coat’s surface properties, including wettability and permeability, facilitating water uptake but also potentially resulting in oxidative stress and cellular damage if not properly controlled [51].

Notably, combinations such as RSC with 2.0 s/cm plasma exposure resulted in the highest percentage of ASs and dead seeds; this reflects the compounded stress from oxidative overexposure. Excessive ROS accumulation in seeds damages the embryonic axis and meristematic cells, reducing seed viability [51,52].

Collectively, the data suggest that the seed coat functions as a physical and biochemical mediator of plasma responsiveness. In intact seeds, it absorbs and regulates plasma-generated species, creating favorable microenvironments for metabolic stimulation. In NHP, controlled exposure enhances hydration but risks localized oxidative effects, while the RSC’s lack of structural buffering renders seeds highly vulnerable to stress, transforming the plasma from a stimulant to a stressor.

Therefore, the efficacy of PT is highly dependent on the SC. Optimization of plasma parameters, including exposure time, intensity, and gas composition, should account for the width and density of tissues in the hilum region. This region exhibits increased permeability to plasma, rendering it more susceptible to internal structural damage. Such damage frequently manifests as morphological abnormalities in the seedling root system (Table 3) (Figure 3b). These abnormalities can serve as practical indicators of the threshold at which plasma conditions become excessively intense for safe and effective seed treatment. Specifically, excessive plasma exposure may compromise seed quality rather than enhance it. Thus, to ensure the appropriate application of plasma, it is crucial to consider the structural integrity of both the seed coat and the hilum. For seeds characterized by thinner seed coats and hilum structures, lower power intensity and shorter exposure durations are recommended to minimize the risk of damage while maintaining treatment efficacy. Although plasma seed treatment can enhance germination and reduce surface-borne pathogens, it may also result in thinner seed coats, increasing susceptibility to pathogen invasion. Improper application can damage the seed coat, compromise defense mechanisms, and further elevate the risk of infection [53]. Therefore, the use of protective fungicides is recommended to prevent subsequent infections [54].

## 4. Materials and Methods

### 4.1. Seed Material and Preparation

Seeds of pepper (*Capsicum* spp.) were provided by a commercial seed company in Chiang Mai, Thailand. The SCs were divided into three conditions: NMS, NHP, and RSC, as shown in Figure 4.

### 4.2. Characteristics of the DBD Plasma Device

Figure 5a illustrates the setup of the DBD plasma device used in this study. The system comprises a rectangular plasma chamber measuring 3 × 25 × 5.8 cm^3^. The electrode gap was 0.8 cm to accommodate the seed size and optimize the plasma exposure. The system operates as a radio frequency transmitter, providing stable output at 13 MHz and 800 W, which is crucial for consistent plasma generation. A high-voltage direct current of 2500 V and 1 A was connected to the electrode terminals. The electrodes consist of stainless-steel plates: a high-voltage electrode (E⁺), which is insulated with a 2 mm thick quartz glass sheet to prevent arcing, and a ground electrode (E−), which serves as the seed-holding substrate for plasma discharge. A gas inlet system ensures a continuous and controlled flow of argon into the chamber at a rate of 6.5 ± 0.5 L/min while maintaining stable plasma conditions. To prevent heat accumulation and protect the electrodes from thermal damage, the air-cooling system was integrated by connecting air-cooled electrodes around the electrode housing. The system circulates cooling air, maintaining a constant temperature during plasma operation. The ground electrode (E−) is mounted on a gear-driven motor system that transports seeds through the plasma field. Adjustable ground plates that move at speeds of 0.4, 0.8, 1.2, 1.6, and 2.0 S/cm ensure uniform treatment and controlled exposure.

### 4.3. Experimental Setup

The experimental design was a split plot with three replications, totaling 150 seeds per treatment (50 seeds per replication). The primary plot factor was the SC (NMS, NHP, and RSC), while the subplot factor was the exposed time passed through the DBD plasma chamber at speeds of 0.4, 0.8, 1.2, 1.6, and 2.0 s/cm and control.

### 4.4. PT

The seeds were arranged in parallel alignment along the *z*-axis at x-positions of 0, 3, 6, and 9 cm, with ten seeds placed in each row, as illustrated in Figure 5b. Plasma treatment (PT) was administered at a fixed position 5 mm below the electrode (y = 5 mm) (Figure 5b), using a power input of 1500 V. The treatment durations were varied at intervals of 0.4, 0.8, 1.2, 1.6, and 2.0 s/cm

Plasma exposure time represents the movement speeds of the ground electrode plate carrying the seeds through the plasma region (1 cm width). These values indicate the time (in seconds), which is calculated as follows:(1)$$speeds\;(s/cm)=\frac{The\;duration\;of\;seed\;displacement\;(s)}{Distance\;traveled\;by\;the\;ground\;electrode\;plate\;(cm)}$$

### 4.5. OES Measurement

The OES served as an effective method for characterizing the reactive species produced within DBD plasma. A wide-spectrum spectrometer (Avaspec-ULS3648 Starline, Avantes, Apeldoorn, The Netherlands), with a spectral range of 200–1100 nm, was employed for OES measurements. To ensure plasma stability and consistent free radical emission, measurements were repeated three times and the results were averaged. Subsequent analysis of the emitted light intensity at each wavelength was performed using Avasoft-Basic version 8.0 (Analytical Software, Amsterdam, The Netherlands) to identify the specific radical species generated.

### 4.6. Effect of Plasma Treatment to Pepper Seed Qualities

#### 4.6.1. Biochemical Test for Viability

The TZ test was pre-conditioned for moistened germination using the blotter method and incubated for 18 h at 20 °C in a seed germination incubator. The seeds were cut the length of seed and transferred to a bottle glass (10 mL), immersed in a tetrazolium chloride (2,3,5-triphenyl tetrazolium chloride) solution 1%, and incubated for 6 h at 30 °C in darkness. After staining, the seeds were rinsed in running water for 1 min [55,56,57] and evaluated based on the following criteria: (1) Viable and vigorous. The embryo and endosperm exhibited uniform pink to dark red coloration, with turgid, firm tissues and no visible lesions (Figure 6a). (2) Viable and non-vigorous are less than 50% of the CDC, turgid, and firm tissue. The radicle–hypocotyl apex may be colorless (Figure 6b). (3) Non-viable. Over 50% of the cotyledon and/or endosperm tissue at the radicle axis exhibited unstained or colorless, flaccid white areas. The endosperm and/or embryo may also be completely devoid of color (Figure 6c) [58].

#### 4.6.2. Germination Tests (Gs)

Gs were performed in three replicates with 50 seeds per replicate using the between method (BP). Seeds were incubated in darkness at 20 °C [59]. Seedlings were counted after sowing on days 7 and 14 wer evaluated based on their morphological characteristics, including normal seedling (Figure 7b), ASs (Figure 3), hard seeds, fresh seeds, and dead seeds [56,57]. The germination percentage was calculated as follows:(2)$$G\left(\%\right)=\frac{number\;of\;normal\;seedlings}{number\;of\;total\;seeds\;sowing}\times 100$$

#### 4.6.3. EC Test

The EC test was conducted on three replicates of 50 seeds per replicate, which were soaked in 50 mL of distilled water and incubated in a growth chamber at 25 °C for 1 h [56]. The EC of the soaking solution was measured and recorded in μS cm^−1^ g^−1^ [59] using a conductivity meter model 914 pH (Metrohm AG, Herisau, Switzerland).(3)$$EC=\frac{conductivity\;reading\left(\mu S\;cm-1\right)-background\;reading}{weight\;of\;replication\;(g)}$$

#### 4.6.4. RE Test

The RE test was counted as radicle emergence (2 mm length from root collar to root tip) after 72 h for seed sowing [57].(4)$$RE\left(\%\right)=\frac{number\;of\;germinated\;seeds\;with\;a\;radicle\;length\;exceeding\;2\;mm}{total\;number\;of\;seeds\;sown}\times 100$$

#### 4.6.5. GI Test

The GI test was counted of NMS daily until 14 days; it was calculated as follows:(5)$$GI=\sum \frac{total\;of\;germination\;seedling\;on\;day}{number\;of\;days\;after\;sowing}$$

#### 4.6.6. SGT Test

After 14 days of sowing, we measured shoot length (hypocotyl) and root length (radicle) (Figure 7a).(6)SGT=(Total of length of shoot/root)/(Number of sample seeds)

### 4.7. Statistical Analysis

The data were analyzed using one-way analysis of variance with the LSD test for multiple comparisons. The statistical significance was determined at the 0.05 level using Statistix version 8.0 (Analytical Software, Tallahassee, FL, USA).

## 5. Conclusions

The results of this study underscore the significant interaction between the SC and PT. The presence and integrity of the seed coat are crucial for determining the efficacy and safety of plasma exposure. PT most effectively enhanced germination and seedling vigor in NMS seed coats. NHP seed coats initially promoted early germination. However, when combined with PT, they exhibited an increase in AS development. In contrast, plasma exposure to RSC seeds was associated with detrimental effects on the overall seed performance.

These findings underscore the importance of customizing PT parameters to align with the specific structural characteristics of seed coats. Beyond serving as physical barriers, seed coats also act as regulatory filters that influence the physiological responses of seeds to external stimuli. Morphological abnormalities in the seedling root, hypocotyl, and cotyledon systems serve as practical indicators for determining the threshold beyond which plasma conditions become excessively intense, thereby compromising both safety and efficacy. For seeds with thinner seed coats and hilum structures, lower power intensities and shorter exposure durations are recommended to minimize the risk of damage while preserving treatment efficacy. Future research should aim to optimize the plasma exposure duration and intensity based on seed morphology and explore the potential integration of pre-treatment methods to enhance seed performance while minimizing stress-related damage.

## Figures and Tables

**Figure 1 plants-14-01938-f001:**
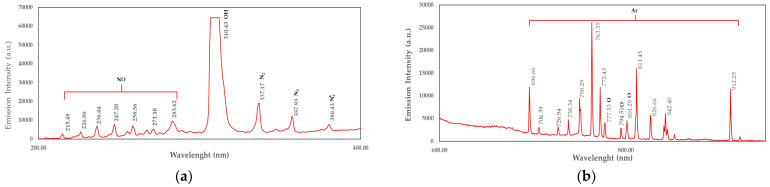
Optical emission spectra for DBD plasma discharge: (**a**) 200–400 nm; (**b**) 600–1000 nm.

**Figure 2 plants-14-01938-f002:**
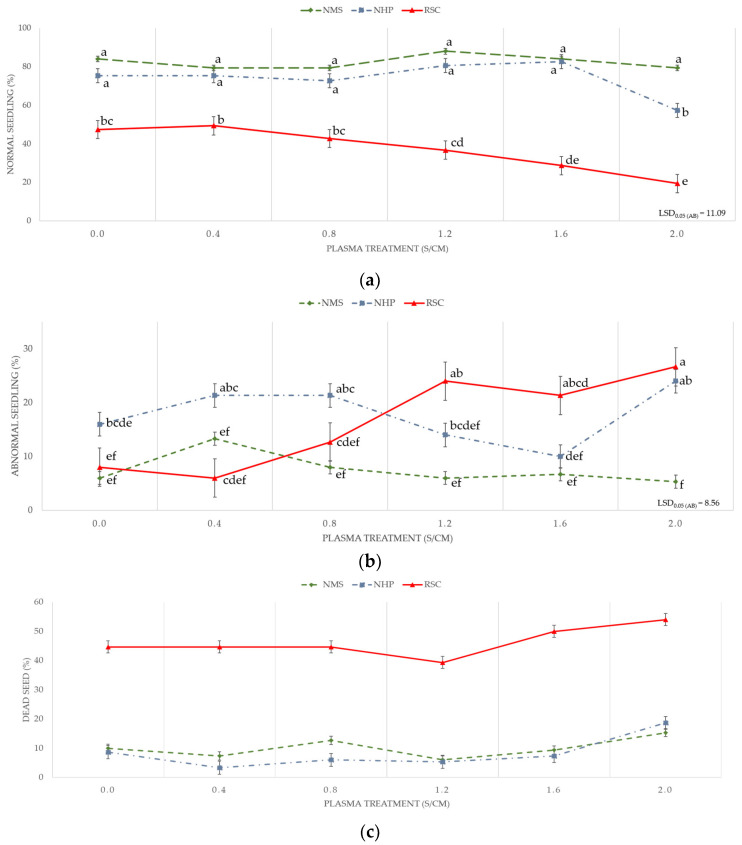
Seed germination rate of pepper seeds at 14 days based on seedling characteristics. (**a**) Normal seedling, (**b**) AS, and (**c**) dead seed after PT at speeds of 0.4, 0.8, 1.2, 1.6, and 2.0 s/cm and control in each seed coat condition. Means with different letters indicate a significant difference between groups (*p* < 0.05), according to the least significant difference (LSD) test.

**Figure 3 plants-14-01938-f003:**
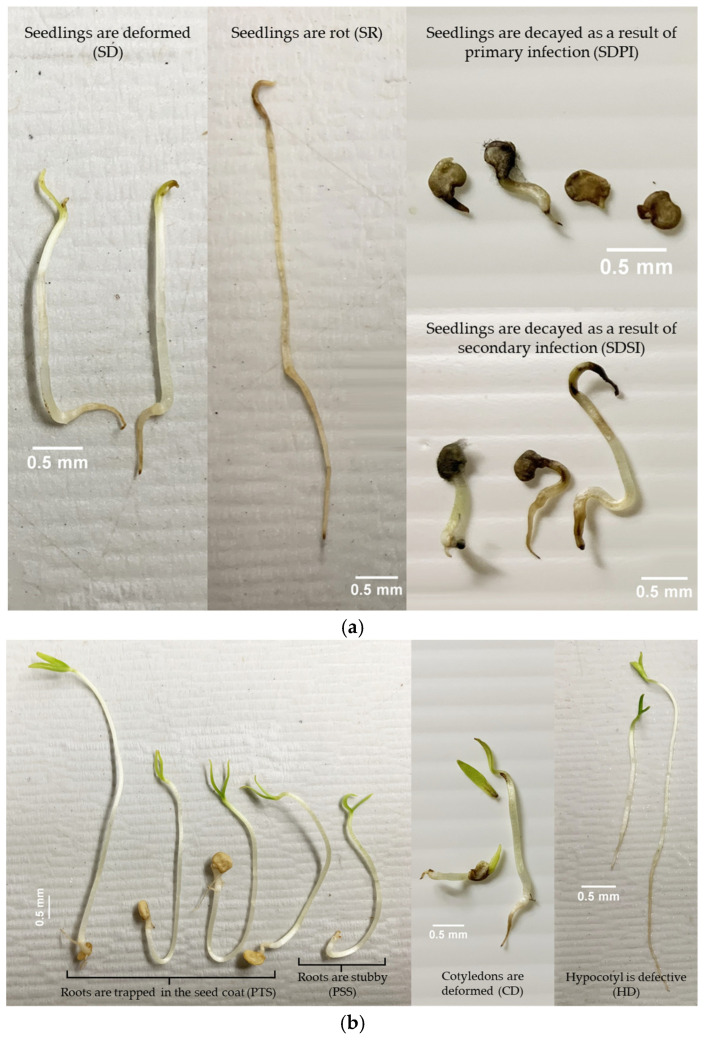
Characteristics of abnormalities in pepper seedlings. (**a**) AS in part of the seedling as a whole; (**b**) AS in part of the root system (**left**) and shoot system (**right**).

**Figure 4 plants-14-01938-f004:**
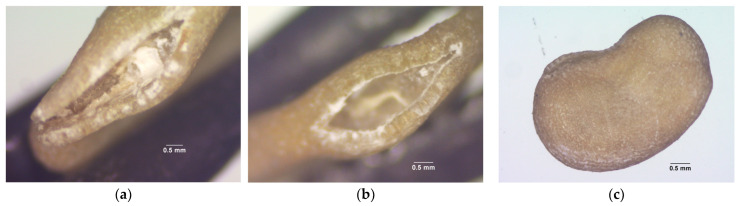
Stereo microscope images (at 30×) of SC for PT. (**a**) NMS; (**b**) NHP; and (**c**) RSC.

**Figure 5 plants-14-01938-f005:**
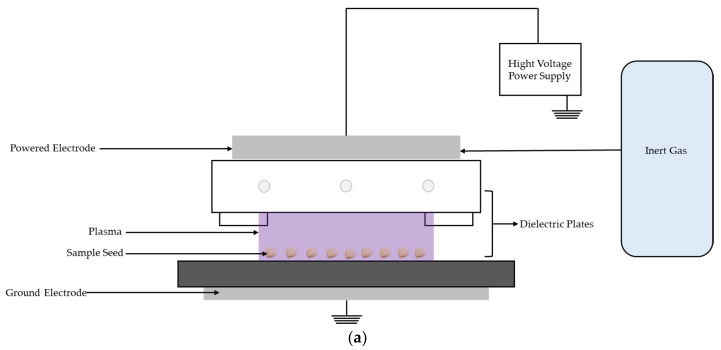
Schematic diagram of (**a**) the DBD plasma devices used for seed treatment; (**b**) the experiment set up with a chamber.

**Figure 6 plants-14-01938-f006:**
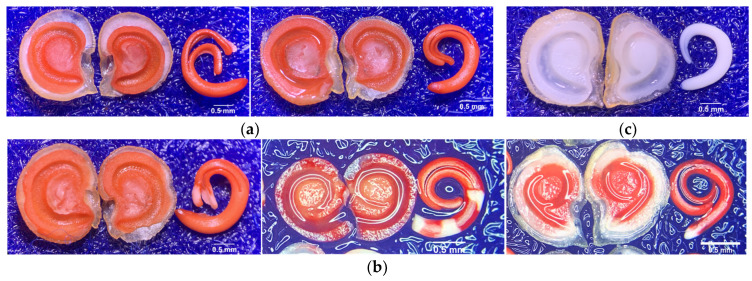
Stereo microscope images (at 20×) of staining pattern of pepper seed. (**a**) Viable and vigorous (VG) embryo and endosperm exhibited uniform pink to dark red coloration; (**b**) viable and non-vigorous (NVG) where the cotyledons (**left**), hypocotyls (**center**), and radicles (**right**) are colorless. (**c**) Non-viable.

**Figure 7 plants-14-01938-f007:**
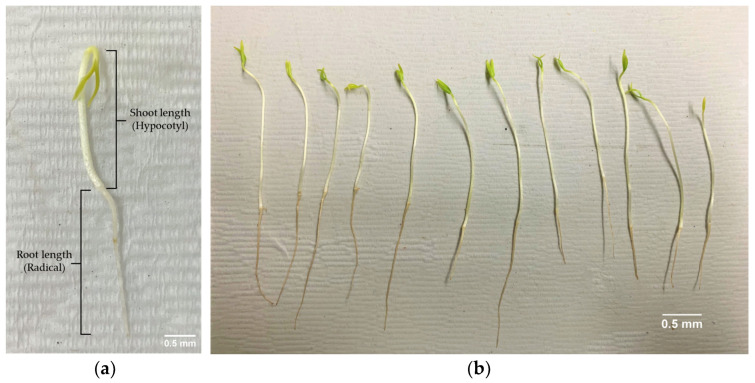
(**a**) Characteristics of normal seedling and measurement of shoot and root lengths; (**b**) germination of pepper seeds 14 days after sowing.

**Table 1 plants-14-01938-t001:** Effect of SC and PT on the viability of pepper seeds.

Factor	Quality Testing				
	TZ (%)				
	Viable				Non-Viable
	VG	NVG			
		CDC	HDC	RDC	
**SC (A)**					
NMS	97 ^a^	0.22 ^b^	0.00 ^b^	0.22 ^b^	2 ^c^
NHP	95 ^b^	0.00 ^b^	0.00 ^b^	0.44 ^ab^	4 ^a^
RSC	94 ^c^	0.66 ^a^	1.44 ^a^	0.56 ^a^	3 ^b^
F-test	**	**	**	*	**
**PT (B)**					
control	99 ^a^	0.00 ^b^	0.00 ^c^	0.00 ^c^	1 ^d^
0.4 s/cm	96 ^bc^	0.56 ^a^	0.00 ^c^	0.89 ^ab^	2 ^c^
0.8 s/cm	97 ^b^	0.00 ^b^	0.67 ^b^	0.44 ^bc^	2 ^c^
1.2 s/cm	97 ^b^	0.00 ^b^	0.00 ^c^	0.00 ^c^	3 ^b^
1.6 s/cm	95 ^c^	0.00 ^b^	1.11 ^a^	0.00 ^c^	4 ^b^
2.0 s/cm	88 ^d^	0.33 ^b^	1.11 ^a^	1.11 ^a^	9 ^a^
F-test	**	**	**	**	**
A × B	**	**	**	**	**
CV (%)	1.18	91.86	71.50	136.91	29.34
LSD_0.05(A)_	0.90	0.25	0.50	0.25	0.33
LSD_0.05(B)_	1.08	0.26	0.33	0.54	0.48

Data are expressed as means, and different letters within the columns indicate significant differences according to the least significant difference (LSD) test. * Significant at *p* ≤ 0.05 and ** *p* ≤ 0.01 level, TZ = tetrazolium test, VG = viable and vigorous, NVG = viable and non-vigorous, CDC = cotyledons displayed colorless, HDC = hypocotyl displayed colorless, and RDC = radicle displayed colorless.

**Table 2 plants-14-01938-t002:** Pepper SC after treatment with DBD plasma and control.

Factor	Quality Testing							
	Germination (G) (%)		Seed Vigor					
	G^1st^	G^Final^	EC (μS cm^−1^ g^−1^)	RE (%)	GI		SGT (cm)	
					7th	14th	Root	Shoot
**SC (A)**								
NMS	55 ^a^	82 ^a^	673.88 ^a^	55.56 ^a^	5.45 ^b^	28.14 ^a^	2.09 ^b^	3.67 ^a^
NHP	55 ^a^	74 ^a^	741.48 ^a^	55.33 ^a^	7.56 ^a^	29.42 ^a^	2.31 ^a^	3.59 ^a^
RSC	15 ^b^	37 ^b^	256.77 ^b^	23.00 ^b^	1.70 ^c^	11.84 ^b^	2.09 ^b^	2.71 ^b^
F-test	**	**	**	**	**	**	*	**
**PT (B)**								
control	40 ^bc^	69 ^a^	518.13 ^bc^	42.44 ^c^	4.93 ^ab^	23.67 ^b^	2.81 ^a^	3.39 ^ab^
0.4 s/cm	41 ^b^	68 ^a^	465.18 ^c^	41.78 ^c^	5.27 ^ab^	23.49 ^b^	2.42 ^b^	3.51 ^a^
0.8 s/cm	42 ^b^	65 ^a^	536.54 ^bc^	41.78 ^c^	5.14 ^ab^	23.53 ^b^	2.19 ^b^	3.53 ^a^
1.2 s/cm	50 ^a^	68 ^a^	622.12 ^a^	54.00 ^a^	5.78 ^a^	26.03 ^a^	1.83 ^c^	3.16 ^bc^
1.6 s/cm	44 ^b^	65 ^a^	619.34 ^a^	48.00 ^b^	4.72 ^b^	23.46 ^b^	2.17 ^b^	3.32 ^abc^
2.0 s/cm	35 ^c^	52 ^b^	582.96 ^ab^	39.78 ^c^	3.58 ^c^	18.65 ^c^	1.58 ^c^	3.02 ^c^
F-test	**	**	**	**	**	**	**	*
A × B	**	**	ns	**	*	**	**	**
CV (%)	12.58	10.30	14.67	10.95	21.11	7.94	12.92	9.36
LSD_0.05(A)_	6.10	12.52	100.78	7.48	1.20	4.00	0.17	0.39
LSD_0.05(B)_	5.07	6.40	78.72	4.71	1.00	1.77	0.27	0.30

Data are expressed as means, and different letters within the columns indicate significant differences according to the LSD test. ns = non-significant; * Significant at *p* ≤ 0.05 and ** *p* ≤ 0.01 level, EC = electrical conductivity, RE = radical emergence, GI = germination index, SGT = seedling growth, SC = seed coat conditions, and PT = plasma treatment.

**Table 3 plants-14-01938-t003:** Abnormal seedlings and dead seeds conditions after DBD PT and control.

Factor	Total AS	Abnormal Seedlings Type (AS)									Dead
		Seedling as a Whole				Root System			Shoot System		
		SD	SDPI	SDSI	SR	PSS	PTS	PM	CD	HD	
**SC (A)**											
NMS	8	0.11	0.00	0.00 ^b^	0.00 ^b^	6.33	1.00 ^b^	0.11	0.00	0.00	10 ^b^
NHP	18	0.67	0.33	0.44 ^b^	1.22 ^a^	5.56	7.44 ^a^	2.00	0.00	0.11	8 ^b^
RSC	16	1.00	0.00	2.56 ^a^	0.11 ^b^	8.00	3.00 ^ab^	0.78	0.33	0.67	46 ^a^
F-test	ns	ns	ns	**	**	ns	*	ns	ns	ns	**
**PT (B)**											
control	10 ^c^	0.00	0.00	1.11	0.00 ^b^	3.33	5.33	0.00 ^b^	0.22	0.00	21 ^b^
0.4 s/cm	14 ^b^	0.44	0.00	0.00	0.00 ^b^	7.56	5.33	0.00 ^b^	0.00	0.22	18 ^b^
0.8 s/cm	14 ^ab^	0.44	0.22	0.67	0.22 ^b^	7.33	4.22	0.67 ^b^	0.22	0.00	21 ^b^
1.2 s/cm	15 ^ab^	0.44	0.44	1.56	0.00 ^b^	8.67	3.33	0.00 ^b^	0.00	0.22	17 ^b^
1.6 s/cm	13 ^b^	1.56	0.00	0.44	0.22 ^b^	5.56	2.22	2.00 ^a^	0.22	0.44	22 ^b^
2.0 s/cm	19 ^a^	0.67	0.00	0.74	2.22 ^a^	7.33	2.44	3.11 ^a^	0.00	0.67	29 ^a^
F-test	*	ns	ns	ns	**	ns	ns	**	ns	ns	**
A × B	**	ns	ns	ns	**	**	ns	**	ns	ns	ns
CV (%)	36.86	235.09	502.00	240.68	109.54	55.41	82.65	133.77	424.26	369.67	30.93
LSD_0.05(A)_	-	-	-	1.15	0.31	-	5.17	-	-	-	8.32
LSD_0.05(B)_	4.94	-	-	-	0.47	-	-	1.24	-	-	6.41

Data are expressed as means, and different letters within the columns indicate significant differences according to the LSD test. ns = non-significant; * Significant at *p* ≤ 0.05 and; ** *p* ≤ 0.01 level. AS = abnormal seedling, SD = seedling is deformed, SDPI = seedling is decayed as a result of primary infection, SDSI = seedling is decayed as a result of secondary infection, SR = seedling is rotten, PSS = primary root is stubby, PTS = primary root is trapped in the seed coat, PM = primary root is missing, CD = cotyledons are deformed, and HD = hypocotyl is defective.

## Data Availability

Data are contained within the article.

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
