# Peer review of "Influence of Seed Coat Integrity on the Response of Pepper Seeds to Dielectric Barrier Discharge Plasma Treatment"

_plants, 2025, doi:10.3390/plants14131938_

Round 1
Reviewer 1 Report
Comments and Suggestions for Authors
The manuscript Influence of Seed Coat Integrity on the Response of Pepper Seeds to Dielectric Barrier Discharge Plasma Treatment is a study aiming to describe the effect of DBD plasma pre-treatment of paper seeds on their germination and viability. The research topic chosen is undoubtedly relevant, up to date and interesting. The manuscript text is chaotically structured, making it difficult to read and understand. The excessively long sentences and many abbreviations used also contribute to this.
My personal impressions and advice to the authors are:
On lines 18-22 of the abstract part, it remains unclear for which of the investigated variants the presented results are.
At the beginning of the introduction there seems to be missing text before the first sentence „These technologies include seed priming, seed stratification, and seed scarification.“ Also the references do not start at number 1 in the next sentence.
I would like to advise the authors to rewrite the definition of plasma in the introduction. At the beginning of the text to omit the definition “Plasma is the fourth state of matter”, as it is controversial and a subject of disagreement among plasma physicists. Due to the lack of a phase transition between the gas state and plasma, the more generally accepted definition is “Plasma is an ionised gas consisting of ions, excited partials, free radicals and electrons that emits ultraviolet light”.
On lines 51 to 53 “These properties make plasma suitable for emerging as a promising alternative to traditional seed conditioning methods. Plasma offers a range of benefits, including improved germination rate, enhanced seedling vigor, and increased resistance to environmental stresses.” it will be better to replace “Plasma” with “Plasma treatment”.
On line 72 “the higher germination rate was at a value of 96.67% compared to the control treatment [17]” the control value is missing.
The sentence on lines 74 to 77 is unclear, need to be rewritten.
There is a lack of motivation for the presentation of optical emission spectra of plasma. The OES presented provide significant information about RONS present in plasma. Their effect on the seeds need to be discussed. On lines 372-373 this discussion is unfinished.
Please explain the speeds of 0.4, 0.8, 1.2, 1.6, and 2.0 s/cm. Is it time for the seeds to settle in the plasma region or is it the speed of movement of the plate?
Author Response
Comments 1: On lines 18-22 of the abstract part, it remains unclear for which of the investigated variants the presented results are. |
Response 1: Thank you for your comment. We agree that the information in lines 18–22 of the abstract was unclear regarding which treatment variants the results referred to. We have revised this section to clearly indicate the specific variants associated with each result. This has now been revised to: “The results indicate that NMS seeds exhibit the highest performance in terms of seed viability. The NMS and NHP had statistically significant higher seed germination, electrical conductivity, radical emergence, germination index at 14 days after sowing and measuring the shoot length than RSC.” (In line 18-22)
|
Comments 2: [At the beginning of the introduction there seems to be missing text before the first sentence „These technologies include seed priming, seed stratification, and seed scarification.“ Also the references do not start at number 1 in the next sentence.]
|
Response 2: Thank you for your comment. We have reviewed the introduction and found that the first part of the sentence was unintentionally omitted during editing. We have revised the text to include the missing context and ensure clarity. Additionally, the reference numbering has been corrected to start from number 1. “Peppers are an essential ingredient in several dishes worldwide, and Thailand’s seed industry holds a prominent position on the world stage, ranking 15th globally and as the third-largest vegetable seed exporter in Asia [1]. The country is known for producing high-quality pepper seeds for export to various countries worldwide. However, the pepper seed export industry faces several challenges, with seed quality being a crucial factor that directly impacts market access, crop yields, and overall competitiveness. Therefore, maintaining seed quality throughout the export process is important for ensuring that the seeds meet international standards. One such practice is a seed enhancement technology, which aims to improve and enhance seed quality after harvesting.” In line 33-41
|
Comments 3: I would like to advise the authors to rewrite the definition of plasma in the introduction. At the beginning of the text to omit the definition “Plasma is the fourth state of matter”, as it is controversial and a subject of disagreement among plasma physicists. Due to the lack of a phase transition between the gas state and plasma, the more generally accepted definition is “Plasma is an ionised gas consisting of ions, excited partials, free radicals and electrons that emits ultraviolet light”. |
Response 3: Thank you for your valuable comment. We agree with your suggestion and have revised the definition of plasma in the introduction accordingly. We removed the phrase “Plasma is the fourth state of matter” and replaced it with the more widely accepted definition: “Plasma is an ionized gas composed of ions, excited particles, free radicals, and electrons. A key characteristic of plasma is its ability to emit electromagnetic radiation, including ultraviolet (UV) light. This emission occurs due to the de-excitation of excited species within the plasma and the recombination of ions and electrons [7].” (In line 57-60)
|
Comments 4: On lines 51 to 53 “These properties make plasma suitable for emerging as a promising alternative to traditional seed conditioning methods. Plasma offers a range of benefits, including improved germination rate, enhanced seedling vigor, and increased resistance to environmental stresses.” it will be better to replace “Plasma” with “Plasma treatment”. |
Response 4: Thank you for your suggestion. We agree with the reviewer’s comment and have replaced “Plasma” with “Plasma treatment” in lines 51 to 53 accordingly. “These properties make plasma treatment suitable for emerging as a promising alternative to traditional seed conditioning methods. Plasma treatment offers a range of benefits, including improved germination rate, enhanced seedling vigor, and increased resistance to environmental stresses.” (In line 60-64)
|
Comments 5: On line 72 “the higher germination rate was at a value of 96.67% compared to the control treatment [17]” the control value is missing. |
Response 5: Thank you for pointing this out. We have added the control value to provide a complete comparison. “DBD PT at 15 minutes with a power of 5.5 kV can be improved; the higher germination rate was at a value of 96.67% compared to the control treatment (86.00%) [18].” (In line 81-83)
|
Comments 6: The sentence on lines 74 to 77 is unclear, need to be rewritten. |
Response 6: Thank you for your comment. We have revised the sentence on lines 74 to 77 to improve clarity. The original sentence was: because DBD plasma in black gram seeds led to an enhancement of nitrogen complex formation in the seed coat. This enhancement promoted nitrogen conversion, resulting in increased protein levels within the seed, which in turn contributed to improved seed germination rates and vigorous seedling growth [25]. Moreover, plasma has the ability to enhance seed germination by stimulating biochemical processes within the seed [26]. This has now been revised to: “DBD plasma treatment of black gram seeds enhanced the formation of nitrogen complexes in the seed coat. The promoted nitrogen conversion increases protein content in the seed coat, enhancing germination rates and seed vigor. This likely results from interactions with reactive plasma species during discharge, which also accelerate imbibition. Nitrogen complexes accumulate by diffusing and absorbing onto the rough, waxy seed coat, where they become partially trapped. As water uptake rises, reactive nitrogen species adsorb onto the uneven seed coat surface and diffuse through its thin layers, further increasing nitrogen content in both the seed and seed coat [26]. Moreover, plasma treatment may stimulate biochemical pathways critical for seed germination. During this process, key biochemical activities are activated, resulting in elevated levels of gibberellic acid (GA₃) and hydrolytic enzymes that facilitate the mobilization of stored nutrients in the endosperm [27].” to improve clarity and ensure the intended meaning is clear to the reader. (In line 83-94)
|
Comments 7: There is a lack of motivation for the presentation of optical emission spectra of plasma. The OES presented provide significant information about RONS present in plasma. Their effect on the seeds need to be discussed. On lines 372-373 this discussion is unfinished. |
Response 7: Thank you for your valuable comment. We agree that the motivation and discussion regarding the OES and their relation to the RONS were insufficient. We have revised the manuscript to clearly explain the significance of the OES data, the role of RONS generated in plasma. This has now been revised to: “The OES served as an effective method for characterizing the reactive species produced within DBD plasma. A wide-spectrum spectrometer (Avaspec-ULS3648 Starline, Avantes, Netherlands), with a spectral range of 200-1100 nm, was employed for OES measurements. To ensure plasma stability and consistent free radical emission, measurements were repeated three times and the results averaged. Subsequent analysis of the emitted light intensity at each wavelength was performed using Avasoft-Basic version 8.0 (Analytical Software, Netherlands) to identify the specific radical species generated.” (In line 409-415) Additionally, the incomplete section on lines were intended as a section heading for testing and were mistakenly placed due to a paragraph formatting error. We have corrected this by moving the heading “4.6 Effect of plasma treatment to pepper seed qualities” to line 416 |
Comments 8: Please explain the speeds of 0.4, 0.8, 1.2, 1.6, and 2.0 s/cm. Is it time for the seeds to settle in the plasma region or is it the speed of movement of the plate? |
Response 8: Thank you for your comment. We have revised the sentence on lines 405 to 406 “Plasma exposure time represents the movement speeds of the ground electrode plate carrying the seeds through the plasma region (1 cm width) These values indicate the time (in seconds).” And we have revised the text to include the missing context and ensure clarity. This has now been revised to: “4.4 PT The seeds were arranged in parallel alignment along the z-axis at x-positions of 0, 3, 6, and 9 cm, with ten seeds placed in each row, as illustrated in Figure 6b. Plasma treatment (PT) was administered at a fixed position 5 mm below the electrode (y = 5 mm) (Figure 6b), using a power input of 1500 V. The treatment durations were varied at intervals of 0.4, 0.8, 1.2, 1.6, and 2.0 s/cm” (In line 398-403)
|
Reviewer 2 Report
Comments and Suggestions for Authors
The aim of the study is briefly mentioned, but the hypothesis is missing. Please clearly state your hypothesis and expected outcome for each seed coat treatment.
Line 123–125-You mention significant differences in viable and non-vigorous seed percentages. Include the exact p-values or statistical test results to support these claims.
Line 159–161: Germination Index Values-Clarify whether the observed differences in GI between treatments were statistically significant, and if so, provide p-values or indicate them in the text.
Line 203–204: Effect of PT on RSC-You describe a "steady decline" in germination for RSC seeds with increased PT exposure. Consider adding statistical support to quantify this trend.
Line 263–266: EC Interpretation-The interpretation of EC as indicative of membrane activity versus damage is not clear. Please expand on the physiological meaning of high EC in viable seeds.
Line 314–317: Vulnerability of RSC-This section suggests that RSC seeds have higher moisture and reduced rigidity. Please reference any supporting literature or provide a rationale for this statement.
Line 331–334: Tailoring PT Parameters-Consider adding a sentence on how your findings could inform the design of commercial plasma treatment protocols or seed quality assessment frameworks.
Consider including key numerical values (e.g., maximum improvement in germination percentage, EC values) in the abstract to strengthen its impact.
The sample size and replication per treatment should be clarified (e.g., how many seeds per treatment, per replication).
Include more discussion on how these results might affect seed treatment recommendations in agricultural practice, particularly in tropical climates.
The conclusion should offer practical guidance for selecting appropriate PT levels for different seed types, and highlight limitations for future research.
Comments on the Quality of English LanguageSeveral sentences are awkward or unclear. Need a thorough language check to improve readability.
Author Response
Comments 1: The aim of the study is briefly mentioned, but the hypothesis is missing. Please clearly state your hypothesis and expected outcome for each seed coat treatment. |
Response 1: Thank you for your valuable comment. We have revised the manuscript to clearly state the hypothesis and the expected outcomes associated with each seed coat treatment. The hypothesis is that different seed coat treatments will result in distinct physiological or germination responses, depending on the nature and intensity of the treatment. This has now been revised to: “However, research focusing on the physical properties of seeds subjected to PT remains limited. Therefore, further investigation is necessary to elucidate the mechanisms underlying seed quality improvement. It is hypothesized that the structure of the seed coat influences the internal seed part response to plasma treatment, which, in turn, affects seed germination and seedling development.”
|
Comments 2: Line 123–125-You mention significant differences in viable and non-vigorous seed percentages. Include the exact p-values or statistical test results to support these claims. |
Response 2: Thank you for your comment. We have revised the manuscript to include the exact p-values and statistical test results for the differences in viable and non-vigorous seed percentages in Lines 133-136. “It significantly decreased to 94% (p≤ 0.05), whereas the VG of NMS was 97%, and the NHP was 95%. Moreover, the results of viable and non-vigorous (NVG) patterns showed that RSC had a statistic significantly increased percentage of NVG compared with NMS and NHP” (In line 141-144)
|
Comments 3: Line 159–161: Germination Index Values-Clarify whether the observed differences in GI between treatments were statistically significant, and if so, provide p-values or indicate them in the text. |
Response 3: Thank you for your comment. We appreciate the reviewer’s comment. The differences in Germination Index (GI) between treatments were statistically analyzed using ANOVA. The results showed that the differences were statistically significant. We have now clarified this in the revised manuscript and included the corresponding p-values in the text (Line 168-170). This has now been revised to: “The NHP seeds exhibited statistically a significant higher germination index (GI) (p ≤ 0.05), with the values recorded at 7 and 14 days after sowing (7.56 and 29.42, respectively). In contrast, the RSC seeds showed the GI values were 1.70 and 11.84, respectively, indicating slower and less uniform germination.” (In line 176-179)
|
Comments 4: Line 203–204: Effect of PT on RSC-You describe a "steady decline" in germination for RSC seeds with increased PT exposure. Consider adding statistical support to quantify this trend. |
Response 4: Thank you for your comment. We have revised the section accordingly. Statistical analysis has been added to support the trend of decreasing germination in RSC seeds with increased plasma treatment (PT) duration. Specifically, ANOVA analysis was performed, and the results are now included in the revised manuscript (Lines 219-220). This has now been revised to: “PT had a negative impact on RSC seedlings, as the percentage of normal seedlings significantly decreased with overexposure time of 1.2 s/cm (Figure 2a).”
|
Comments 5: Line 263–266: EC Interpretation-The interpretation of EC as indicative of membrane activity versus damage is not clear. Please expand on the physiological meaning of high EC in viable seeds. |
Response 5: Thank you for your comment. We have clarified the interpretation of EC in the manuscript. “In contrast to prolonged exposure time, plasma can disrupt seed cell membranes, leading to the leakage of intracellular compounds and a consequent increase in the EC of the surrounding water. This elevated EC serves as an indicator of the role seed vigor, excessive EC resulting from overexposure can hinder germination and growth. [29].” The revised text now elaborates on this interpretation for better clarity. (In line 280-284)
|
Comments 6: Line 314–317: Vulnerability of RSC-This section suggests that RSC seeds have higher moisture and reduced rigidity. Please reference any supporting literature or provide a rationale for this statement. |
Response 6: Thank you for your comment. The statement regarding reduced rigidity of RSC seeds is based on our experimental results, as shown in table 2 and figure 2. The higher moisture content may contribute to reduced mechanical strength, as also suggested by Waskow et al., 2021 and Starič et al., 2020. We have revised the section to clarify this and added a proper citation. And We have revised the section to include supporting literature. This has now been revised to: “The RSC showed heightened sensitivity to PT, as evidenced by the increased rates of AS (Figure 2b) and dead seeds (Figure 2c). This heightened sensitivity can be attributed to the inherent physical and biochemical properties of RSC seeds, such as higher moisture content [13] and reduced structural rigidity, which render them more susceptible to plasma induced damage [50].” (In line 333-336)
|
Comments 7: Line 331–334: Tailoring PT Parameters-Consider adding a sentence on how your findings could inform the design of commercial plasma treatment protocols or seed quality assessment frameworks. |
Response 7: Thank you for the suggestion. We have added a sentence to highlight how our findings could be applied to the development of commercial plasma treatment protocols and seed quality assessment frameworks (Lines 349-360) This has now been revised to: “Therefore, the efficacy of PT is highly dependent on the SC. Optimization of plasma parameters, including exposure time, intensity, and gas composition, should account for the width and density of tissues in the hilum region. This region exhibits increased permeability to plasma, rendering it more susceptible to internal structural damage. Such damage frequently manifests as morphological abnormalities in the seedling root system (Table 3) (figure 4b). These abnormalities can serve as practical indicators of the threshold at which plasma conditions become excessively intense for safe and effective seed treatment. Specifically, excessive plasma exposure may com-promise seed quality rather than enhance it. Thus, to ensure the appropriate application of plasma, it is crucial to consider the structural integrity of both the seed coat and the hilum. For seeds characterized by thinner seed coats and hilum structures, lower power intensity and shorter exposure durations are recommended to minimize the risk of damage while maintaining treatment efficacy.”
|
Comments 8: Consider including key numerical values (e.g., maximum improvement in germination percentage, EC values) in the abstract to strengthen its impact. |
Response 8: Thank you for the suggestion. While we understand the importance of including numerical values, due to word limitations in the abstract, we have instead emphasized the trends and key findings. However, these values are clearly presented in the results section. This has now been revised to: “The results indicate that NMS seeds exhibit the highest performance in terms of seed viability. The NMS and NHP had statistically significant higher seed germination, electrical conductivity, radical emergence, germination index at 14 days after sowing and measuring the shoot length than RSC. Plasma exposure at 1.2 s/cm improved germination and vigor, whereas 2.0 s/cm exposure significantly decreased seed viability and increased the number of abnormal seedlings. The interaction between SC and PT significantly affected seedling abnormalities, with RSC seeds being more vulnerable to damage under prolonged exposure.” (In line 18-25)
|
Comments 9: The sample size and replication per treatment should be clarified (e.g., how many seeds per treatment, per replication). |
Response 9: We used 150 seeds per treatment and each treatment was replicated 3 times, with 50 seeds per replication. This information has been added to the Materials and Methods section for clarity in Line 393-397. This has now been revised to: “4.3 Experimental Setup The experimental design was a split plot with three replications, totaling 150 seeds per treatment (50 seeds per replication). The primary plot factor was the SC (NMS, NHP, and RSC), while the subplot factor was the exposed time passed through the DBD plasma chamber at speeds of 0.4, 0.8, 1.2, 1.6, and 2.0 s/cm and control.”
|
Comments 10: Include more discussion on how these results might affect seed treatment recommendations in agricultural practice, particularly in tropical climates. |
Response 10: Thank you for your valuable comment. We have added a discussion on the potential implications of our findings for agricultural seed treatment practices in tropical climates. “Therefore, the efficacy of PT is highly dependent on the SC. Optimization of plasma parameters, including exposure time, intensity, and gas composition, should account for the width and density of tissues in the hilum region. This region exhibits increased permeability to plasma, rendering it more susceptible to internal structural damage. Such damage frequently manifests as morphological abnormalities in the seedling root system (Table 3) (figure 4b). These abnormalities can serve as practical indicators of the threshold at which plasma conditions become excessively intense for safe and effective seed treatment. Specifically, excessive plasma exposure may compromise seed quality rather than enhance it. Thus, to ensure the appropriate application of plasma, it is crucial to consider the structural integrity of both the seed coat and the hilum. For seeds characterized by thinner seed coats and hilum structures, lower power intensity and shorter exposure durations are recommended to minimize the risk of damage while maintaining treatment efficacy. Although plasma seed treatment can enhance germination and reduce surface-borne pathogens, it may also result in thinner seed coats, increasing susceptibility to pathogen invasion. Improper application can damage the seed coat, compromise defense mechanisms, and further elevate the risk of infection [52]. Therefore, the use of protective fungicides is recommended to prevent subsequent infections [53]. (In line 349-365)
|
Comments 11: The conclusion should offer practical guidance for selecting appropriate PT levels for different seed types, and highlight limitations for future research. |
Response 11: Thank you for your valuable suggestion. We have revised the conclusion to include practical recommendations for selecting suitable plasma treatment levels for different seed types. In addition, we have highlighted the limitations of the current study and proposed directions for future research. This has now been revised to: “The results of this study underscore the significant interaction between the SC and PT. The presence and integrity of the seed coat are crucial for determining the efficacy and safety of plasma exposure. PT most effectively enhanced germination and seedling vigor in NMS seed coats. NHP seed coats initially promoted early germination. However, when combined with PT, they exhibited an increase in AS development. In contrast, plasma exposure to RSC seeds was associated with detrimental effects on the overall seed performance. These findings underscore the importance of customizing PT parameters to align with the specific structural characteristics of seed coats. Beyond serving as physical barriers, seed coats also act as regulatory filters that influence the physiological responses of seeds to external stimuli. Morphological abnormalities in the seedling root, hypocotyl, and cotyledon systems serve as practical indicators for determining the threshold beyond which plasma conditions become excessively intense, thereby compromising both safety and efficacy. For seeds with thinner seed coats and hilum structures, lower power intensities and shorter exposure durations are recommended to minimize the risk of damage while preserving treatment efficacy. Future research should aim to optimize the plasma exposure duration and intensity based on seed morphology and explore the potential integration of pre-treatment methods to en-hance seed performance while minimizing stress-related damage.” (In line 455-473) |
4. Response to Comments on the Quality of English Language |
Point 1: Several sentences are awkward or unclear. Need a thorough language check to improve readability. |
Response 1: Thank you for your valuable feedback. We have carefully revised the manuscript for language clarity and improved the structure of awkward or unclear sentences to enhance overall readability. A thorough language check has been conducted throughout the manuscript. Additionally, the manuscript has been proofreading by a native English editor from Language School and Translation Services. (https://www.languagecorner.ac.th/) |

Reviewer 3 Report
Comments and Suggestions for Authors
The research addresses an important technological question: how the integrity of the seed coat influences pepper seeds' response to dielectric barrier discharge plasma treatment (PT), particularly in terms of seed viability, germination, and seedling vigor.
The topic is moderately relevant and partially original within the domain of agricultural plasma applications. It addresses an acknowledged gap in understanding the interaction between seed coat integrity and plasma treatment, a promising area with implications for sustainable seed enhancement. However, it mostly reiterates known plasma-plant interactions rather than delivering substantially new insights. This is not a groundbreaking contribution since similar studies in other crops already highlight seed coat–plasma interactions, limiting the novelty of these findings.
The conclusions generally align with the presented data, affirming the hypothesis that seed coat integrity significantly influences plasma treatment outcomes. Nonetheless, the conclusions are too generic and somewhat overstated considering the limitations in the data, particularly the lack of direct mechanistic measurements like reactive oxygen species, so they should be presented with more caution.
The references are generally appropriate and up to date.
The tables and figures are generally informative and clear, presenting key trends and quantitative data. However, some figures lack error bars (Fig.2) or detailed captions explaining experimental conditions, which would help readers interpret variability and significance.
In addition, the authors should consider including more detailed morphological analyses (e.g., microscopic imaging) of seed coat damage. In particular, it remains unclear how all the pictures (Fig. 5; what about the scale to understand the geometry) were taken. Is it optical microscopy? Seed surface morphology should be studied at different magnifications using scanning electron microscopy before and after exposure to PT to illustrate surface changes.
The PT experimental conditions were described insufficiently. How many seeds were used? Was it a single experiment or repeated a few times? What was the temperature during PT procedures? etc.
It is written that to prevent heat accumulation and protect the electrodes from thermal damage, an air-cooling mechanism was integrated into the system. How was it done?
Optical emission spectroscopy was done; however, what equipment was used?
Author Response
Comments 1: The research addresses an important technological question: how the integrity of the seed coat influences pepper seeds' response to dielectric barrier discharge plasma treatment (PT), particularly in terms of seed viability, germination, and seedling vigor. |
Response 1: Thank you for highlighting the importance of our research objective. We appreciate your recognition of our efforts to investigate the role of seed coat integrity in response to dielectric barrier discharge plasma treatment, particularly its impact on seed viability, germination, and seedling vigor.
|
Comments 2: The topic is moderately relevant and partially original within the domain of agricultural plasma applications. It addresses an acknowledged gap in understanding the interaction between seed coat integrity and plasma treatment, a promising area with implications for sustainable seed enhancement. However, it mostly reiterates known plasma-plant interactions rather than delivering substantially new insights. This is not a groundbreaking contribution since similar studies in other crops already highlight seed coat–plasma interactions, limiting the novelty of these findings. |
Response 2: Thank you for the comment. We agree that plasma–seed interactions have been studied before, but our work focuses specifically on pepper seeds, which are less explored. We believe the findings still add useful information, especially regarding seed coat integrity in this species. We’ve adjusted the discussion to better reflect this point.
|
Comments 3: The conclusions generally align with the presented data, affirming the hypothesis that seed coat integrity significantly influences plasma treatment outcomes. Nonetheless, the conclusions are too generic and somewhat overstated considering the limitations in the data, particularly the lack of direct mechanistic measurements like reactive oxygen species, so they should be presented with more caution. |
Response 3: Thank you very much for your valuable feedback. We appreciate your insight regarding the conclusions. We agree that some statements could be more cautiously presented given the limitations of our data. We have revised the conclusion section accordingly to better reflect these limitations and avoid overstating the findings. This has now been revised to: “The results of this study underscore the significant interaction between the SC and PT. The presence and integrity of the seed coat are crucial for determining the efficacy and safety of plasma exposure. PT most effectively enhanced germination and seedling vigor in NMS seed coats. NHP seed coats initially promoted early germination. However, when combined with PT, they exhibited an increase in AS development. In contrast, plasma exposure to RSC seeds was associated with detrimental effects on the overall seed performance. These findings underscore the importance of customizing PT parameters to align with the specific structural characteristics of seed coats. Beyond serving as physical barriers, seed coats also act as regulatory filters that influence the physiological responses of seeds to external stimuli. Morphological abnormalities in the seedling root, hypocotyl, and cotyledon systems serve as practical indicators for determining the threshold beyond which plasma conditions become excessively intense, thereby compromising both safety and efficacy. For seeds with thinner seed coats and hilum structures, lower power intensities and shorter exposure durations are recommended to minimize the risk of damage while preserving treatment efficacy. Future research should aim to optimize the plasma exposure duration and intensity based on seed morphology and explore the potential integration of pre-treatment methods to en-hance seed performance while minimizing stress-related damage.” (In line 445-473)
|
Comments 4: The references are generally appropriate and up to date. |
Response 4: Thank you for your kind comment on the references.
|
Comments 5: The tables and figures are generally informative and clear, presenting key trends and quantitative data. However, some figures lack error bars (Fig.2) or detailed captions explaining experimental conditions, which would help readers interpret variability and significance. |
Response 5: Thank you for your valuable comment. We have revised the figures accordingly. Specifically, we have added error bars to Fig. 2 to reflect data variability and included more detailed captions for all relevant figures to clarify the experimental conditions. These changes aim to improve the clarity and interpretability of the results.
|
Comments 6: In addition, the authors should consider including more detailed morphological analyses (e.g., microscopic imaging) of seed coat damage. In particular, it remains unclear how all the pictures (Fig. 5; what about the scale to understand the geometry) were taken. Is it optical microscopy? Seed surface morphology should be studied at different magnifications using scanning electron microscopy before and after exposure to PT to illustrate surface changes. |
Response 6: Thank you for your valuable suggestion. We acknowledge the importance of detailed morphological analysis. In response, we have revised the figure caption of Fig. 5 to clearly indicate that the images were taken using stereo microscopy and have added scale bars to help understand the geometry. We also included an additional explanation in the Results section regarding the imaging method.
Although scanning electron microscopy (SEM) analysis was not performed in the current study due to equipment limitations, we agree that SEM would provide more detailed insights into seed surface morphology. We will consider incorporating SEM analysis in our future work to further investigate the surface changes induced by plasma treatment.
|
Comments 7: The PT experimental conditions were described insufficiently. How many seeds were used? Was it a single experiment or repeated a few times? What was the temperature during PT procedures? etc. |
Response 7: Thank you for your comment. We have revised the manuscript to clarify the experimental conditions of plasma treatment (PT) as follows: These details have been added to the revised manuscript in the methods section (lines 393-397). This has now been revised to: “The experimental design was a split plot with three replications, totaling 150 seeds per treatment (50 seeds per replication). The primary plot factor was the SC (NMS, NHP, and RSC), while the subplot factor was the exposed time passed through the DBD plasma chamber at speeds of 0.4, 0.8, 1.2, 1.6, and 2.0 s/cm and control.
What was the temperature during PT procedures? etc. Seed and plasma electrode plate temperatures were previously monitored during the plasma treatment (PT) process using a thermal imaging analyzer. The seed temperature was consistently maintained at approximately 30–35 °C, and the temperature of the seeds remained within a safe range. Please find attached the related temperature measurement data for reference. (The figure showed in attached file)
|
Comments 8: It is written that to prevent heat accumulation and protect the electrodes from thermal damage, an air-cooling mechanism was integrated into the system. How was it done? |
Response 8: We appreciate the reviewer’s comment. This has now been revised to: “The air-cooling system was integrated by connecting air-cooled electrodes around the electrode housing. The system circulates cooling air, maintaining a constant temperature during plasma operation.” (In line 385-387)
|
Comments 9: Optical emission spectroscopy was done; however, what equipment was used? |
Response 9: We appreciate the reviewer’s comment. Optical emission spectroscopy was performed using Avaspec-ULS3648 Starline, Avantes, Netherlands. This information has been added to the revised manuscript. This has now been revised to: “The OES served as an effective method for characterizing the reactive species produced within DBD plasma. A wide-spectrum spectrometer (Avaspec-ULS3648 Starline, Avantes, Netherlands), with a spectral range of 200-1100 nm, was employed for OES measurements. To ensure plasma stability and consistent free radical emission, measurements were repeated three times and the results averaged. Subsequent analysis of the emitted light intensity at each wavelength was performed using Avasoft-Basic version 8.0 (Analytical Software, Netherlands) to identify the specific radical species generated. (In line 409-415)
|

Round 2
Reviewer 1 Report
Comments and Suggestions for Authors
I would like to thank the authors for accepting my comments.
Author Response
We sincerely appreciate your valuable comments and suggestions, which have helped us improve the manuscript.
Reviewer 3 Report
Comments and Suggestions for Authors
The manuscript was improved following most of the reviewers' comments.
Just a minor remark: scaling is added to Fig.5; the same should be done to Figs. 3, 4, and 7.
Author Response
Comments 1: Just a minor remark: scaling is added to Fig.5; the same should be done to Figs. 3, 4, and 7. |
Response 1: Thank you for your suggestion. We have added the scaling to Figures 3, 4, and 7 accordingly. |